# Exploring the Neuroprotective Potential of *Astragalus membranaceus* in Central Nervous System Diseases

**DOI:** 10.3390/biom15121671

**Published:** 2025-12-01

**Authors:** Jiajia Sang, Jialin Gao, Hui Zuo, Haolu Yu, Yuqi Qin, Jun Hu, Feng Hao

**Affiliations:** Nanjing University of Chinese Medicine, Nanjing 210023, China

**Keywords:** *A. membranaceus*, bioactive compounds, central nervous system diseases, mechanism of action

## Abstract

*Astragalus membranaceus (A. membranaceus)*, a traditional Chinese medicine, has gained increasing recognition for its potential in treating central nervous system (CNS) disorders. This review aims to systematically integrate the mechanisms of action of *A. membranaceus* and its bioactive compounds on CNS diseases, with a focus on exploring its therapeutic potential and introducing related health food products. We conducted a comprehensive literature search in PubMed and Web of Science from January 2015 through July 2025. Our analysis reveals that *A. membranaceus* and its bioactive compounds, particularly *A. membranaceus* IV (AS-IV) and *A. membranaceus* polysaccharides (APS), exert multifaceted neuroprotective effects. These effects encompass the mitigation of neuroinflammation, oxidative stress, apoptosis, and ferroptosis, as well as the regulation of autophagy and protection of the blood–brain barrier. The therapeutic potential of *A. membranaceus* is linked to the modulation of key signaling pathways, such as NF-κB, Nrf2, and PI3K/Akt. Furthermore, based on the concept of “homology of medicine and food,” *A. membranaceus* is being developed into various health food formulations, offering a promising strategy for the adjuvant treatment and preventive care of CNS diseases. In conclusion, *A. membranaceus* represents a promising, multi-target pharmacological agent for CNS disorders, yet further high-quality clinical studies are warranted to validate its efficacy and safety in humans.

## 1. Introduction

Central Nervous System (CNS) disorders continue to climb up the global burden of disease list, and it has been reported that the number of deaths in 2019 alone is as high as 10 million [1]. For instance, the occurrence of neurodegenerative diseases like Parkinson’s disease (PD) continues to rise annually, leading to patient distress [2,3]. The severity of CNS disease is not only reflected in the direct destruction of neurological function, but also in the chain reaction of multiple systems. For instance, hepatic encephalopathy, a severe issue associated with advanced liver disease, can quickly result in reduced consciousness, coma, and potentially death [4]. These conditions not only endanger individuals’ lives and health but also impose a significant burden on society and families due to their high rates of disability and long-term care needs [5]. Hence, patients with CNS disorders are in urgent need of novel therapeutic approaches. Given their origin from natural resources and low toxicity, traditional Chinese medicines (TCM) have gradually gained wide recognition in the preventive and therapeutic fields.

*Astragalus membranaceus* (*A. membranaceus*), the dried root of *A. membranaceus* (Fisch.) Bge. var. Mongholicus (Bge.) [6] is renowned as “the holy medicine” for enhancing qi. First mentioned in Shennong Ben Cao Jing [7], it has a long history of medicinal use and is one of the most commonly used TCM for CNS disorders (Figure 1) [8]. Recent pharmacological research indicates that *A. membranaceus* offers therapeutic benefits through multiple targets and pathways for preventing and treating CNS diseases, due to its diverse bioactive compounds. Until now, researchers have identified more than 200 compounds from *A. membranaceus*, comprising polysaccharides, saponins, flavonoids, alkaloids, among others, which form a treasure trove of structurally diverse and functionally rich natural small molecules [9]. Among them, *A. membranaceus* IV (AS-IV) and *A. membranaceus* polysaccharides (APS) have attracted the most attention because of their outstanding biological activities, which together constitute the “main force” of *A. membranaceus* in exerting therapeutic and pharmacological effects [6]. AS-IV, a form of AST, is vital for its anti-inflammatory and neuroprotective properties by inhibiting NF-κB and other pathways [10]. Cycloastragenol (CAG) is a potent derivative of AS-IV, known for its ability to reduce inflammation and improve lipid metabolism [11]. APS is a water-soluble, chemically complex heteropolysaccharide that comprises portions of heteropolysaccharides, glucans, and both neutral and acidic polysaccharides, and it has notable effects on modulating immune cell activity [12,13]. These representative active ingredients work together to carry the scientific basis for the overall efficacy of *A. membranaceus*.

Currently, the body of research concerning *A. membranaceus* remains relatively sparse, with a notable absence of systematic reviews addressing its role and potential mechanisms in CNS disorders. Consequently, this study seeks to conduct a comprehensive review of the active constituents of *A. membranaceus* and elucidate their molecular mechanisms in the context of CNS disease treatment. The findings aim to furnish novel insights and references for future investigations into *A. membranaceus* and to inform the development of innovative strategies for the prevention and management of CNS disorders.

## 2. Methods

### 2.1. Literature Search

PubMed and Web of Science were searched for reports on the effects of *A. membranaceus* and its components on CNS diseases using the following search keywords: “*A. membranaceus*”, “Astragali Radix”, “AS-IV”, “APS”, “CNS diseases”, and “pharmacology”. The search period spanned from January 2015 to July 2025, with a focus on incorporating high-impact studies from the past decade.

### 2.2. Inclusion Criteria

The inclusion criteria encompassed studies focusing on *A. membranaceus* and its active constituents in relation to CNS disorders, including AD, TBI, PD, and SCI, as well as pharmacological investigations of *A. membranaceus*.

### 2.3. Exclusion Criteria

Studies that were duplicates, incomplete, lacked ethical approval, were presented as conference abstracts, appeared as brief newsletters, or were published in languages other than English were excluded.

## 3. Results

Two researchers independently executed the search strategy using the specified keywords and screened titles, abstracts, and full texts against the inclusion and exclusion criteria, resulting in 218 eligible publications. The literature search and screening process is summarized in Figure 2.

## 4. Impact of *A. membranaceus* and Its Bioactive Components on the CNS

*A. membranaceus* and its bioactive constituents exert curative effects on CNS diseases via multiple pathways and targets (Table 1).

### 4.1. Anti-Neuroinflammation

Neuroinflammation is a key pathogenetic mechanism in many CNS diseases, such as cerebral infarction, spinal cord injury (SCI), and Alzheimer’s disease (AD). It has long since ceased to be a “bystander” to CNS disease and is one of the central mechanisms driving disease onset, progression, and determining prognosis [31]. One of the primary causes of neuroinflammation is the activation of microglia [32]. When tissues are inflamed, macrophages of the activated M1 phenotype secrete inflammatory and chemotactic proteins to help the host resist the infection and then transform into the activated M2 phenotype to repair the damaged tissues [33,34]. Nuclear factor-κB (NF-κB) acts as a key transcriptional regulator in the inflammatory response [35]. Upon activation, it enhances the secretion of pro-inflammatory molecules, including IL-6, IL-8, TNF-α, IL-1β, COX-2, and iNOS, thereby intensifying the inflammatory cascade [36]. NLRP3, a coreceptor in inflammatory vesicles, is linked to inflammation in various CNS disorders and can trigger local and systemic inflammatory responses. Inhibiting NLRP3 can significantly reduce inflammation and protect the CNS (Figure 3) [37,38].

#### 4.1.1. Regulatory Effects of *A. membranaceus* Components (AS-IV, APS) on Microglia/Macrophages

Suppressing microglial overactivation represents a promising therapeutic strategy for various neurological disorders. AS-IV has been identified as a possible neuroprotective and anti-inflammatory compound for CNS disorders. In a mouse model of cerebral ischemia, AS-IV demonstrated anti-inflammatory properties by converting microglia/macrophages from the M1 to the M2 phenotype through a mechanism dependent on peroxisome proliferator-activated receptor γ (PPARγ) [14,39]. It simultaneously decreases pro-inflammatory elements like IL-1β, IL-6, TNF-α, MyD88, NF-κB, and TLR4, while boosting anti-inflammatory components such as ARG-1, CD206, and IL-10 [15,40]. APS also reverses M1/M2 polarization, promotes ATP degradation to release anti-inflammatory factors, and inhibits P2X7R expression, providing neuroprotection in the cerebral cortex of MCAO rats [24,41].

#### 4.1.2. Inhibitory Effect of AS-IV on NLRP3 Inflammatory Body and Related Damage

NLRP3 levels were notably increased in brain tissues affected by hypoxic–ischemic conditions, and its overexpression exacerbated the degree of brain tissue damage. Inhibition of its activity became an important way to protect the CNS [42]. In the rat MCAO/R model, AS-IV decreased LOC102555978 expression to lessen cerebral infarction volume and suppress inflammation and cell death [16]. Furthermore, in a mouse model of depression caused by repeated restraint stress (RRS) and LPS, AS-IV administration significantly increased GSK3β phosphorylation and inhibited NLRP3 inflammatory vesicles, and decreased the levels of inflammatory factors, such as TNF-α and IL-1β [17]. The combination of AS-IV and hydroxy safflower yellow A treatment in ischemic brain showed a significant increase in NLRP3/ASC/IL-1β/Caspase-1/1/GSDMD protein expression, which was significantly more pronounced, indicating that the combination suppressed NLRP3 inflammasome-mediated cellular pyroptosis, reduced the inflammatory response, and mitigated brain tissue damage [43].

#### 4.1.3. Regulatory Effect of *A. membranaceus* on Astrocytes

Astrocytes are the “igniters, amplifiers, and regulators” of the immune cascade in a variety of human CNS diseases [44]. Research conducted in vitro has demonstrated that activated astrocytes produce high levels of inflammatory mediators, including IL-1β and TNF-α [45,46]. AS-IV significantly inhibited penicillin-induced expression of inflammatory factors and p-MAPK family proteins in astrocytes, and significantly alleviated neurological damage in epileptic mice [47]. AS-IV also inhibited the senescence of astrocytes and prevented dopaminergic neuronal damage in PD by promoting mitochondrial autophagy, thereby avoiding the degeneration of dopaminergic neurons in PD and demonstrating significant therapeutic potential [48]. APS protects astrocytes from OGD/R-induced neuroinflammation to a certain extent by blocking the HMGB1/RAGE/NF-κB/NLRP3 signaling pathway [49].

#### 4.1.4. Regulatory Effect of *A. membranaceus* on Neutrophils

Neutrophils are integral components of the innate immune system, serving as the initial responders to sites of tissue injury or pathogen invasion, where they participate in the inflammatory response [50]. This characteristic underlies their extensive infiltration and activation within brain tissue during the inflammatory response to cerebral ischemia/reperfusion. Consequently, inhibiting this pathway may provide adequate protection against CNS diseases [51]. AS-IV mitigates the accumulation of neutrophils in the brain parenchyma by reducing the concentration of MPOs in brain tissue, the percentage of CD11b/CD18-positive neutrophils, and neutrophil-associated molecules in brain tissue up to 24 h post-reperfusion, thereby enhancing neurological outcomes. This reduction in accumulation within the brain parenchyma contributes to improved neurological outcomes and a decrease in infarct volume [52].

#### 4.1.5. Regulatory Effects of *A. membranaceus* Through Multiple Inflammatory Signaling Pathways

Inflammatory signaling pathways play a central role in the pathogenesis and progression of neurological disorders. In the context of traumatic brain injury (TBI), AS-IV significantly contributes to minimizing neuroinflammation and brain damage by downregulating the expression of inflammatory markers, including IL-6, IL-1β, and TNF-α, through the PERK pathway [18]. Moreover, addressing PERK-mediated stress in the endoplasmic reticulum reduces neuroinflammation and improves depressive-like symptoms [53]. In the PC12 neuronal inflammation model induced by 1 mg/mL LPS, pretreatment with AS-IV significantly reduced TNF-α, IL-1β, and TLR4 levels and inhibited the IL-17 signaling pathway [54]. APS also suppresses TNF-α, IL-1β, and TLR4 through the NF-κB and MAPK (ERK, JNK) pathways [55].

Although numerous preclinical studies have demonstrated the potent anti-neuroinflammatory effects of AS-IV and APS in regulating microglial polarization, inhibiting NLRP3 inflammasome, and regulating astrocyte activation, these findings, derived primarily from animal models and in vitro experiments, require validation in human studies. For instance, AS-IV demonstrated the ability to transform from an M1 to an M2 phenotype in the MCAO model [14], but its immunomodulatory effect in human ischemic stroke remains unclear. In addition, the regulatory results of pathways such as NF-κB and NLRP3 in various studies exhibit specific heterogeneity, suggesting that multiple factors, including model type, administration timing, and dose, may influence their effects. Future studies on human cells and clinical samples are needed to clarify their precise role in the complex neuroinflammatory network.

### 4.2. Oxidative Stress

Oxidative stress is characterized by the disruption of cellular mitochondria and the endoplasmic reticulum due to damaging stimuli, leading to the excessive production of reactive oxygen species (ROS) and reactive nitrogen species (RNS) [56]. This disruption disturbs the balance between oxidative and antioxidant systems, resulting in the accumulation of oxidative products and a decrease in reductase activity. This series of events ultimately triggers the peroxidation of DNA/RNA, proteins, and lipids, as well as tissue damage, culminating in a pathological state [57,58,59]. Numerous studies have demonstrated that ROS levels are significantly elevated in the pathophysiology of conditions such as glioma, AIS, AD, and TBI, among others. Common types of ROS include H_2_O_2_, -OH, and O_2_^−^ [60]. Maintaining moderate levels of reactive oxygen species during neuronal development is essential for executing critical physiological functions and participating in complex signaling processes (Figure 3) [61].

#### 4.2.1. *A. membranaceus* Regulates the Expression Levels of Antioxidant Enzymes

In the CNS, superoxide dismutase (SOD) and glutathione peroxidase (GSH-Px) are crucial, with SOD converting superoxide anion to H_2_O_2_ and GSH-Px further converting H_2_O_2_ to H_2_O and O_2_, thereby scavenging ROS and free radicals and attenuating CNS oxidative stress damage. Increasingly accumulating evidence suggests that AS-IV can improve the operations of SOD and GSH-Px while reducing the accumulation of oxidative products, such as ROS, in CNS disease models, including ischemia–reperfusion and experimental autoimmune encephalomyelitis, thereby exerting its biological effects against lipid peroxidation [59,62,63,64,65,66]. In addition to the above results, AS-IV reduced the expression of NADPH oxidase 2/4 (NOX2/4) and increased the total antioxidant capacity (T-AOC) in the brain tissues of Acute Ischemic Stroke (AIS) mice [67]. Calycosin (CA), a flavonoid component of *A. membranaceus*, showed potential neuroprotective effects by decreasing the expression of MDA, NO, and LDH and inhibiting oxidative stress in a SCI model in combination with rehabilitation training [27].

#### 4.2.2. The Regulation of Mitochondrial Function by *A. membranaceus*

Abnormal mitochondrial function is regarded as a central causative factor of oxidative stress [68]. An intrinsic key factor driving a series of complex pathological alterations, which play a pivotal role in the process of neurological injury, ionic homeostasis imbalance, and impaired nerve regeneration [69]. *A. membranaceus* and its active ingredients have been shown to have significant potential to modulate various mitochondrial functions. AS-IV regulates mitochondrial permeability transition, upregulates Bc1-2 levels, inhibits Caspase-3 activation, and attenuates neurological damage caused by AD, among others [70]. The extract from *A. membranaceus* root can counteract oxidative damage in the brain tissue of epileptic mice caused by pentylenetetrazole (PTZ) by enhancing mitochondrial complex activity and membrane potential, thereby providing anticonvulsant effects [71].

#### 4.2.3. Nrf2 Plays a Core Role in the Antioxidant Stress Resistance of *A. membranaceus*

Nrf2 is a key transcription factor that regulates the expression of genes encoding antioxidant enzymes, which are essential for cellular defense against oxidative stress. It binds to Keap1, enters the nucleus, and activates genes like HO-1 and SOD [72,73]. Activating Nrf2 may offer new treatments for neurological disorders. AS-IV shields cortical neurons from OGD/R damage through the activation of EGFR-Nrf2 signaling [74]. AS-IV protects neurons from OGD/R damage via EGFR-Nrf2 signaling and, with Panax ginseng’s (Rg1), activates the Nrf2/HO-1 pathway to combat oxidative stress [75]. Studies suggest that Formononetin (FMN), an isoflavonoid in *A. membranaceus*, offers neuroprotection by increasing Nrf2 expression in rats suffering from TBI, thereby regulating redox homeostasis [76]. Interestingly, the Chinese herbal formula Huangqi Guizhi WuWu Tang (HGWD) reduced oxaliplatin-induced neurotoxicity, primarily by regulating antioxidant precursors and essential molecules in the PI3K/Akt-Nrf2 pathway, which subsequently decreased the oxidative response triggered by paclitaxel in the CNS [77].

*A. membranaceus* and its bioactive components have demonstrated significant antioxidant potential in various CNS disease models by enhancing the activities of antioxidant enzymes such as SOD and GSH-Px, regulating mitochondrial function, and activating the Nrf2 pathway. However, it is not clear whether these effects have the same efficacy in the human body. For example, AS-IV protects cortical neurons in the OGD/R model through EGFR-Nrf2 signaling [74], but whether the regulatory mechanism of this pathway in human neurons is consistent remains to be verified. Moreover, it remains unclear whether long-term or high-dose administration of *A. membranaceus* antioxidants might induce compensatory oxidative stress or disrupt redox homeostasis.

### 4.3. Anti-Apoptosis

Apoptosis is a self-initiated cell death process triggered by normal cells in both physiological and pathological states, playing a vital role in maintaining biological balance [78]. Genes that control apoptosis can be divided into three groups: pro-apoptotic genes (e.g., Fas, Bax, ICE, p53) [79,80], anti-apoptotic genes (e.g., bcl-2, EIB, IAP) [81], and genes with bidirectional regulatory functions (e.g., c-myc) [82]. These genes determine cell fate through a sophisticated network; however, it is the caspase cascade that ultimately carries out programmed cell death, and this pathway is regarded as the common endpoint for the convergence of apoptotic signals [83]. Upon further study, researchers found that apoptosis, as seen in neuronal cells, is at the core of the dramatic decline in neurological function and the rapid deterioration of the disease (Figure 4).

#### 4.3.1. *A. membranaceus* Inhibits JNK Phosphorylation

Activation of c-Jun N-terminal kinase (JNK) leads to the initiation of apoptosis [19]. Inhibition of JNK phosphorylation may be a key pathway for protecting against neuronal apoptosis. AST attenuates neuronal apoptosis in ischemia/reperfusion rats through this mechanism, increasing the expression of phosphorylated extracellular signal-regulated kinase (p-ERK) and phosphorylated Akt [20]. Sun’s team experimentally verified that AS-IV, administered orally for 3 days, in addition to down-regulating p-JNK and upregulating p-ERK, p Akt, in addition to down-regulating p-JNK, upregulating p-ERK, p-ERK, and Klotho, and inhibiting oxidative stress and pro-inflammatory factor release, AS-IV also protects the developing brain from anesthesia-induced neuronal apoptosis [84].

#### 4.3.2. *A. membranaceus* Regulates the Expression of CaSR

In cerebral ischemia–reperfusion injury (CIRI), the calcium-sensitive receptor (CaSR) is an essential G-protein-coupled receptor that exacerbates secondary neurological injury upon activation. Increasing evidence suggests that AS-IV inhibits apoptosis and attenuates neurological deficits secondary to CIRI by down-regulating the expression of CaSR and apoptotic proteins such as Fas, FasL, Bid, and Caspase-8. It also attempts to inhibit cleaved caspase-3 and upregulate the Bax/Bcl-2 ratio, suggesting that the dynamic regulation of CaSR may be one of the targets of AS-IV’s action to exert neuroprotective effects and inhibit apoptosis [85,86]. AS-IV not only reduces apoptotic protein expression but also inhibits endoplasmic reticulum stress-related proteins like eIF2a, Bip, and CHOP, thereby decreasing cerebral infarction by preventing endothelial cell apoptosis [29].

#### 4.3.3. *A. membranaceus* Inhibits Apoptosis Through Other Pathways

Research has demonstrated that AS-IV provides anti-apoptotic benefits in the hippocampus of AD rats, thereby enhancing cognitive function by reducing Bax and caspase-3 concentrations [87]. Wang et al. discovered that AST suppressed key proteins in the Fas/Fasl-VDAC1 pathway in Aβ1-42-damaged C8D1A cells, thereby reducing apoptosis and enhancing cognitive function in hippocampal astrocytes, which helps alleviate cognitive deficits in AD mice [88]. Notably, PPARγ is a key molecular switch for the treatment of AD, and its inhibition reduces AβO-induced apoptosis of HT22 cells by AS-IV [89].

*A. membranaceus* components exert anti-apoptotic effects through mechanisms such as inhibiting JNK phosphorylation, down-regulating CaSR expression, and regulating the Bax/Bcl-2 ratio. These findings have been repeatedly verified in models such as ischemia–reperfusion and AD. However, the apoptotic pathway may play a dual role at different stages of the disease. Excessive inhibition of apoptosis may hinder the necessary cell clearance process. For example, AS-IV alleviates apoptosis in the CIRI model by inhibiting CaSR, but the physiological role of CaSR during the neural repair period has not been fully considered [86]. Furthermore, most studies have focused on a single pathway, lacking a systematic exploration of the interactions between apoptosis and other forms of cell death, such as autophagy and pyroptosis. In the future, it will be necessary to combine multi-omics technologies to reveal the regulatory nodes within the complete biological network.

### 4.4. Autophagy Regulation

Autophagy maintains cellular balance by removing damaged organelles, faulty proteins, and pathogens through lysosomal degradation and recycling [90,91]. Numerous studies have demonstrated that abnormalities in autophagy and endolysosomal pathways, associated with neuronal dysfunction, are intricately linked to the pathogenesis of CNS disorders. Consequently, the regulation of autophagy may play a crucial role in the therapeutic strategies for CNS diseases (Figure 4) [92,93,94].

### A. membranaceus Regulates Autophagy via AMPK and mTOR

Adenosine-activated protein kinase (AMPK) is a crucial energy sensor that helps maintain cellular energy balance [95]. AMPK activity regulates a broad spectrum of metabolic processes, encompassing the modulation of autophagy, mitophagy, and various other mechanisms [96]. Rapamycin, a principal regulator of cellular metabolism [97], exerts its inhibitory effects on autophagy by phosphorylating and inactivating critical regulatory proteins, including ULK1, Beclin-1, UVRAG, and TFEB, as well as by suppressing the expression of autophagy-related proteins [30]. Consequently, AMPK and mTOR are proposed to play pivotal roles in regulating autophagy. Neuronal apoptosis is a major cause of secondary injury after SCI. Research conducted by Lin’s team demonstrated that the intraperitoneal administration of AS-IV can induce autophagy in neuronal cells through the mTORC1 signaling pathway, thereby conferring protection against apoptosis by modulating autophagy. This suggests that AS-IV holds promise as a novel therapeutic agent for SCI [21]. Researchers found that AST plays a key neuroprotective role by stimulating the PI3K/Akt-mTOR pathway-mediated autophagy and upregulating the levels of autophagy flux-associated proteins in APP/PS1 mice, which represents a new mechanism of AST for treating AD through autophagy regulation [98]. Hao et al. demonstrated that AS-IV can attenuate autophagy activity by inhibiting the AMPK/mTOR pathway, thereby reducing mitochondrial-mediated apoptosis. This action mitigates cellular injury in SH-SY5Y cells and the MCAO model, thereby exerting neuroprotective effects and positioning AS-IV as a potential therapeutic candidate for AIS [99]. Zhang and colleagues identified a novel mechanism for treating ischemia/reperfusion (I/R) injury, wherein AS-IV inhibits apoptosis by enhancing P62-LC3-mediated autophagy and increasing the expression of LC3II/LC3I in HT22 cells following oxygen-glucose deprivation/reperfusion (OGD/R) treatment. This suggests that the promotion of autophagy regulation by AS-IV represents a novel therapeutic mechanism for I/R injury [100]. Furthermore, AS-IV modulates the AMPK-MTORC1-ULK pathway to enhance autophagy, as evidenced by reduced P-mTOR levels and increased levels of LC3, P-AMPK, and P-ULK in retinal tissues, thereby exerting a therapeutic effect on traumatic optic neuropathy (TON) [101].

Autophagy has a “double-edged sword” characteristic in CNS diseases, and its regulation requires a high degree of spatiotemporal specificity. For instance, in the SCI model, AS-IV promotes autophagy and reduces apoptosis by inhibiting mTORC1 [21], but in the AIS model, it weakens autophagy to protect neurons by inhibiting AMPK/mTOR [99]. These seemingly contradictory findings suggest that the effects of *A. membranaceus* may be highly context-dependent, varying with the specific disease model or stage. Currently, there is no systematic study comparing the dynamic effects of *A. membranaceus* on autophagic flux at different disease stages, nor has its interaction with processes such as inflammation and oxidative stress been elucidated. In the future, models that more closely resemble human diseases should be constructed to assess the clinical translational potential of autophagy regulation.

### 4.5. Anti-Ferroptosis

Ferroptosis is a regulated form of programmed cell death that results from iron-triggered phospholipid peroxidation [101,102]. Since brain tissue is rich in lipids and iron and has a high rate of oxygen consumption, which makes ferroptosis particularly strong, a defense strategy against ferroptosis may be an important tool in the treatment of CNS diseases (Figure 5) [28].

#### 4.5.1. The Regulation of GPX4 by *A. membranaceus*

Glutathione peroxidase 4 (GPX4) serves a distinctive role as a critical negative regulator of ferroptosis, primarily through its ability to convert lipid hydroperoxides into non-toxic lipid alcohols, thus interrupting the lipid peroxidation chain reaction. The proper functioning of GPX4 is crucial for regulating ferroptosis. Inhibition of GPX4 may increase cellular susceptibility to ferroptosis [22]. Total flavonoids (TFA) extracted from *A. membranaceus* inhibited ferroptosis by upregulating the SLC7A11/GPX-4 axis, as well as by increasing intracellular glutathione levels and GSH/GSSG ratio. Additionally, researchers found that the use of the flavonoids extracted from *A. membranaceus* as a dietary supplement might be beneficial for PD patients [104]. Zhang et al. demonstrated that AS-IV enhanced the sensitivity of SLC7A11 and GPX4, both of which serve as indicators for evaluating ferroptosis. Additionally, AS-IV activated the Nrf2/HO-1 signaling pathway, thereby inhibiting ferroptosis and mitigating neuronal death in the MCAO model by modulating critical nodes within this pathway [105]. Interestingly, AS-IV administration in subarachnoid hemorrhage (SAH) similarly activates this pathway, inhibiting lipid peroxide accumulation and blocking the ferroptosis process in SAH [106]. Mai’s research team created selenium nanoparticles (TSIIA/TMP/APS@Se NPs) containing APS, Tan-shinoneIA, and tetramethylpyrazine, which effectively inhibited ferroptosis. These nanoparticles reversed the decline in neuronal numbers and GPX4 levels after SCI and significantly reduced 4-hydroxynonenal, a marker of lipid peroxidation, potentially improving functional recovery after SCI [107].

#### 4.5.2. *A. membranaceus* Relies on NADPH to Function

The commonality of enzymes and metabolites necessary for resistance to lipid peroxidation during ferroptosis lies in their dependence on the key cellular reducing agent, NADPH [101]. In HT22 cells, AS modulators restored the balance of brain iron metabolism and stabilized lipid peroxidation by reducing NADPH oxidase 4 (NOX4) and activating the NOX4/Nrf2 signaling pathway. Peroxidation (LPO) homeostasis and inhibited ferrocyte apoptosis in SAMP8 mice, a mechanism that suggests the potential of AS modulators to inhibit ferroptosis and improve AD symptoms, making them a potential therapeutic candidate [108].

*A. membranaceus* components inhibit ferroptosis by upregulating GPX4, activating the Nrf2/HO-1 pathway, and regulating NOX4, etc., showing good prospects in models such as SAH, PD, and AD. However, ferroptosis, as a new form of programmed cell death, has not yet been fully elucidated in its molecular mechanism, and it remains unclear whether *A. membranaceus* components have direct targets in this process. For example, AS-IV inhibits ferroptosis in the MCAO model through the Nrf2/HO-1 pathway [105], but whether this pathway is its primary mechanism of action remains controversial. Additionally, there is currently limited research on how the interaction between ferroptosis and other modes of cell death, such as apoptosis and autophagy, changes under the intervention of *A. membranaceus*. In the future, it is necessary to combine gene editing and lipidomics technologies to clarify their specific contributions to the ferroptosis network.

### 4.6. Anti-Blood–Brain Barrier Damage

The blood–brain barrier (BBB) serves as the CNS’s final defense against external threats. Its structural and functional stability is vital for preserving the neuronal environment (Figure 5). However, under pathological conditions such as AIS, AD, and multiple sclerosis (MS), the BBB suffers from structural damage due to inflammatory cascades, oxidative stress imbalance, and metabolic reprogramming, with cascading increases in permeability, which exacerbates the formation of a neurotoxic microenvironment and triggers irreversible neuronal damage and dysfunction [109].

#### 4.6.1. The Regulation of Tight Junction Proteins by Active Ingredients of *A. membranaceus*

AST completely reversed LPS-induced BBB leakage and depressive behavior in a mouse model by down-regulating MMP-9 and upregulating Claudin-5, suggesting that it may be an adjunctive neuroprotectant for the treatment of sepsis-associated encephalopathy (SAE) [23]. AS-IV increases TEER and reduces sodium fluorescein extravasation, while also increasing the expression of compact proteins, such as zonula-1 and occludin, in LPS-stimulated bEnd.3 cells to protect the BBB [26]. Research has demonstrated that lanthanum, serving as a vascular permeability tracer during the intravenous administration of AS-IV, exclusively stains cerebral capillaries. This finding strongly suggests the BBB-protective properties of AS-IV, as it significantly inhibits the upregulation of MMP-9 and AQP4 and mitigates cerebral vasogenic edema [110]. CAG not only mitigated oligomeric Aβ 1-42-induced apoptosis in bEnd.3 cells and enhanced the expression of tight junction scaffolding proteins, but also facilitated the efflux of soluble Aβ across the BBB by upregulating P-glycoprotein (P-gp) and downregulating receptor for advanced glycation end-products (RAGE) expression [103]. Isoastragaloside I (ISOI), a cyclic alkane glycoside derived from *A. membranaceus*, not only restored the reduced levels of tight junction proteins in LPS-stimulated bEnd.3 cells but also reduced BBB permeability by activating the Nrf2 signaling pathway [25]. The combined administration of *A. membranaceus* and Chuanxiongzine diminishes hemorrhage and Evans blue dye extravasation in the brain tissue of I/R mice, while enhancing the expression of tight junction proteins, thereby ameliorating the ultrastructural disruption of the BBB [111].

#### 4.6.2. The Regulation of ETS1 by *A. membranaceus*

The transcription factor ETS1 is the central switch that triggers the endothelial-mesenchymal transition EndoMT and then disrupts the BBB. APS shows potential application in MS prevention by inhibiting the overexpression of ETS1, blocking endoMT, and stabilizing tight junctions [112].

#### 4.6.3. *A. membranaceus* Combats the BBB Through Other Means

*A. membranaceus* injection activates the BDNF/TrkB/CREB pathway, improving BBB function and reducing sepsis-induced neurological deficits [113]. Furthermore, Hou et al. identified that AS-IV effectively inhibits endoplasmic reticulum stress-mediated endothelial cell apoptosis, thereby protecting the BBB and reducing the cerebral infarct area in I/R rats [29].

*A. membranaceus* and its components protect the integrity of the BBB by enhancing the expression of tight junction proteins, inhibiting ETS1, and regulating ER stress, and have shown significant effects in models such as LPS, I/R, and SAE. However, most of these results originated from short-term acute models, and their long-term protective effects in chronic BBB degenerative diseases (such as AD and MS) have not yet been verified. For instance, AS-IV enhances the expression of ZO-1 and occludin in LPS-stimulated bEnd.3 cells [26], but its effect in human brain microvascular endothelial cells and its ability to penetrate the BBB remain to be confirmed. In addition, the distribution, metabolism, and potential cumulative toxicity of *A. membranaceus* components in the BBB destruction area have not been systematically evaluated. In the future, imaging and biomarkers should be combined to promote their translational research from basic to clinical applications.

## 5. Pharmacognosy of *A. membranaceus*

### 5.1. Safety

On 9 November 2023, *A. membranaceus* was officially recognized as an edible Chinese medicinal substance by China’s health authorities. It shows low toxicity with no reported clinical side effects. In a study by Yu’s team, rats and beagles received RAE (APS and AS) for 90 days without significant toxicity, even at doses 70 and 35 times higher than the human dose, respectively [114]. Xie’s team conducted three-month gavage experiments on rats using *A. membranaceus*, noting no fatalities or toxicity, confirming its safety. They established a NOAEL of 8800 mg/kg/day, 50 times the human clinical dose, further verifying the formulation’s safety [115]. In summary, *A. membranaceus* and its main bioactive compounds are safe at standard doses in both short- and long-term toxicity tests, showing no significant side effects. However, high doses or prolonged use may lead to toxicity, so caution is advised.

### 5.2. Application of A. membranaceus in the Field of Food and Medicine

#### 5.2.1. Application of *A. membranaceus* in Formulated Preparations

*A. membranaceus* has been used in treating CNS diseases since its inclusion in Buyang Huanwu Decoction, as noted in Wang Qingren’s “Medical Forest Right and Wrong.” Modern studies have shown that this decoction enhances nerve function and prevents neuronal damage through a multi-target mechanism. It reduces oxidative stress by inhibiting ROS, lowering MDA and 8-OHdG, and boosting SOD and GSH-Px. It also restores mitochondrial function, stabilizes energy metabolism, activates the PKCε/Nrf2/HO-1 pathway, and enhances antioxidant gene expression, forming a protective loop. These effects collectively reduce cerebral I/R injury and support recovery, supporting *A. membranaceus*’s use in treating IS-related qi deficiency and blood stasis [116].

#### 5.2.2. Application of *A. membranaceus* in Health Foods

*A. membranaceus* is now added to health foods like drinks, pastries, and porridges, supporting the “medicine and food” philosophy (Figure 6). Patents show its use in three categories: liquid, semi-liquid, and solid, all for easy consumption (Table 2). These foods can boost the immune system and impact intestinal microbiota, which, in turn, influence both digestion and brain health through the gut–brain axis [117]. Thus, consuming *A. membranaceus*-infused health foods may indirectly benefit brain health.

##### Liquid Products

Soups are common in medicinal diets. For example, *A. membranaceus* chicken soup, made with black-boned chicken and goji berries, boosts qi, nourishes blood, and alleviates sub-health issues such as qi and blood deficiency and fatigue. It can also help with appetite loss in diseases like PD and act as a supportive therapy.

##### Semi-Liquid Products

Semi-liquid products, renowned for their unique texture and ease of digestion, are particularly suitable for patients with CNS diseases following surgery or radiochemotherapy. *A. membranaceus* porridge, combining *A. membranaceus* for kidney and spleen health with Japonica rice for stomach benefits, helps alleviate symptoms like memory loss and joint pain in AD patients with spleen and kidney deficiencies.

##### Solid Products

Within the staple food category, *A. membranaceus* health biscuits are not only effective in alleviating hunger but also contribute to the physical well-being of patients with CNS disorders, offering health-enhancing benefits.

## 6. Summary

### 6.1. Neuroprotective Potential and Mechanisms of A. membranaceus in CNS Disorders

CNS diseases are marked by high levels of morbidity, mortality, and disability, posing a serious threat to human health. However, in current clinical practice, the efficacy of conventional drug therapy is not satisfactory and is accompanied by adverse effects. Natural Chinese medicines derived from herbs or plants offer the advantages of biological activity, safety, and accessibility to resources. They are becoming a valuable treasure trove for the development of new drugs. The significant disability burden associated with CNS diseases is further compounded by the toxicity of many current therapeutics, making the search for low-toxicity, plant-derived neuroprotective agents an urgent need in the field of neurology. *A. membranaceus* has been used in TCM for a long time [118]. Its signature active ingredients exert anti-neuroinflammation, anti-oxidative stress, anti-apoptosis, autophagy regulation, anti-ferroptosis, and anti-BBB damage effects, which deeply intervene in the core mechanism of CNS diseases. Ferroptosis, a recently identified form of programmed cell death, has garnered considerable attention in the context of CNS diseases in recent years. Contemporary research suggests that its principal mechanisms involve signaling pathways such as GPX4, NOX4, and Nrf2. *A. membranaceus* and its active constituents modulate ferroptosis through several mechanisms: they activate the Nrf2/HO-1 pathway, upregulate GPX4 expression, and enhance antioxidant capacity; they also inhibit NOX4 activity, reduce ROS generation, and stabilize lipid peroxidation balance. These effects have been substantiated in various CNS disease models, demonstrating notable neuroprotective properties. As previously discussed, the recent study revealed that *A. membranaceus* exhibits significant neuroprotective effects against a range of CNS diseases through a synergistic mechanism characterized by a “multi-component-multi-target-multi-pathway” model, as demonstrated in both in vivo and in vitro models. Notably, preclinical studies consistently demonstrate that *A. membranaceus* is free from both acute and long-term toxicity, ensuring high safety, and it is listed in the ‘medicine and food’ catalog [119]. In recent years, the idea of “medicinal food” has shifted *A. membranaceus* from “adjuvant treatment” to a focus on “early prevention and functional rehabilitation,” offering a new strategy for precise intervention in CNS diseases.

### 6.2. The Limitations Faced in the Clinical Application and Promotion of A. membranaceus

Though the neuroprotective effects of *A. membranaceus* in CNS diseases have been widely validated, its promotion in clinical practice still faces multiple limitations:

From Single Pathways to Integrative Multi-Omics and Human-Relevant Models. The major hurdle of unclear pathways and targets necessitates a paradigm shift from studying isolated signaling axes to a systems-level, integrative approach. Future work should employ an integrative multi-omics strategy—combining spatial transcriptomics, proteomics, and lipidomics—to profile the comprehensive effects of *A. membranaceus* in relevant disease models [120]. This data must be integrated with network pharmacology to construct a quantifiable interaction network, moving beyond descriptive mechanisms to predictive models that can identify key synergistic nodes. Furthermore, to bridge the species gap and enhance human relevance, the field should leverage advanced human stem cell-derived models, such as brain organoids and blood–brain barrier-on-a-chip systems [121]. These models provide a physiologically accurate platform for validating network predictions and dissecting the complex cell–cell communication mediated by *A. membranaceus*, ultimately translating holistic synergy into quantifiable, precise intervention strategies.

From Empirical Combination to Rational Pharmacokinetic/Pharmacodynamic (PK/PD) Studies. The role of *A. membranaceus* in drug combinations is unclear, with potential risks for altered efficacy or adverse reactions when co-administered with conventional neurotherapeutics. Systematic investigations into pharmacokinetic and pharmacodynamic (PK/PD) interactions are therefore essential. Research should specifically examine how *A. membranaceus* influences the absorption, distribution, metabolism, and excretion of CNS drugs. Utilizing in vitro models, such as Caco-2 cells and human liver microsomes, can determine whether it acts as an inhibitor or inducer of key metabolic enzymes (e.g., CYP450) or transporters (e.g., P-gp) [122]. This will establish a scientific basis for rational combination therapies, maximizing synergy and minimizing clinical risks, a critical step before large-scale clinical adoption.

From Preclinical Validation to Mechanism-Informed Clinical Trial Design. A significant gap exists between robust animal data and the lack of high-quality clinical evidence. Critical parameters, such as dose-exposure relationships and long-term safety in humans, remain undefined. Future research should prioritize well-designed, mechanism-informed clinical trials. Prior to large-scale trials, Phase I/IIa studies are needed to establish human pharmacokinetics and identify preliminary biomarkers of target engagement (e.g., neuroinflammation imaging, plasma neurofilament light levels) [123]. Subsequent multicenter, randomized controlled trials (RCTs) should focus on specific patient populations where preclinical evidence is strongest, such as using AS-IV-standardized extracts as an adjunctive therapy in early Parkinson’s disease or vascular cognitive impairment. Furthermore, long-term observational studies are required to definitively assess the risks of tolerance and other potential side effects associated with prolonged use.

As a result, to successfully transition *A. membranaceus* from the research laboratory setting to practical clinical use, we must gain a more detailed understanding of its underlying mechanisms of action. This involves conducting thorough scientific investigations to uncover precisely how *A. membranaceus* functions within the biological systems. Additionally, it is crucial to refine and optimize dosage protocols to ensure that therapeutic benefits are maximized while minimizing potential side effects. This optimization process should be based on comprehensive pharmacokinetic and pharmacodynamic studies. Furthermore, conducting high-quality, rigorous clinical trials is imperative to establish the safety and efficacy of *A. membranaceus* in various patient populations. These trials should be designed with robust methodologies, including randomized controlled trials, to generate reliable and statistically significant data. These concerted efforts are crucial for establishing the clinical efficacy of *A. membranaceus* and facilitating its translation into standard medical practice.

## 7. Conclusions

In summary, our review consolidates compelling evidence that *A. membranaceus* and its principal bioactive constituents, notably AS-IV and APS, function as a multifaceted pharmacological agent against a spectrum of CNS disorders. Its neuroprotective efficacy is underpinned by a sophisticated “multi-component, multi-target, multi-pathway” mechanism, which orchestrates the concurrent mitigation of neuroinflammation, oxidative stress, apoptosis, and ferroptosis, while also regulating autophagy and fortifying the blood–brain barrier. This polypharmacological profile positions *A. membranaceus* as a particularly attractive candidate for addressing the complex, intertwined pathologies of neurodegenerative and cerebrovascular diseases.

The paradigm of “homology of medicine and food” further elevates its translational potential, enabling a strategic shift from adjuvant therapy towards early prevention and long-term functional rehabilitation through health food products. However, the translation of these robust preclinical findings into clinical practice is contingent upon addressing critical gaps. Future research must pivot from descriptive mechanism elucidation to a predictive, systems-level understanding employing integrative multi-omics and human-relevant disease models. Furthermore, rigorous pharmacokinetic studies and well-designed, mechanism-informed clinical trials are imperative to definitively establish its efficacy, optimal dosing, and long-term safety in humans. By systematically navigating these challenges, *A. membranaceus* can be fully validated as a cornerstone in the evolving arsenal of evidence-based, natural neuroprotective strategies.

## Figures and Tables

**Figure 1 biomolecules-15-01671-f001:**
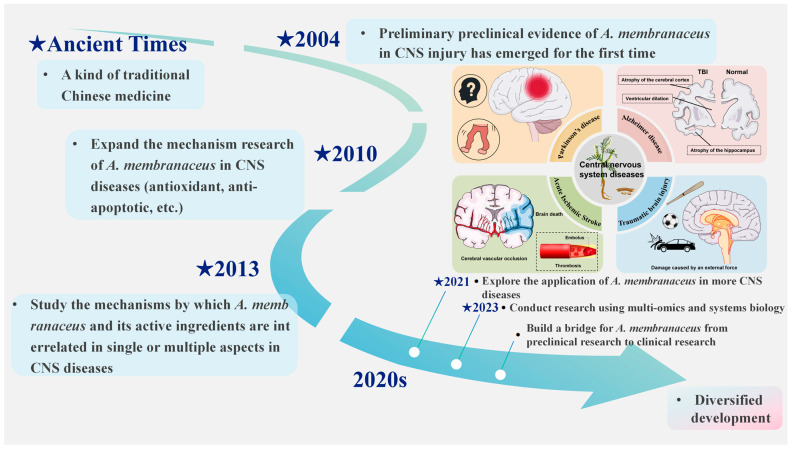
*A. membranaceus* as a treatment option for central nervous system disorders. This figure is original and was created by the authors for this publication.

**Figure 2 biomolecules-15-01671-f002:**
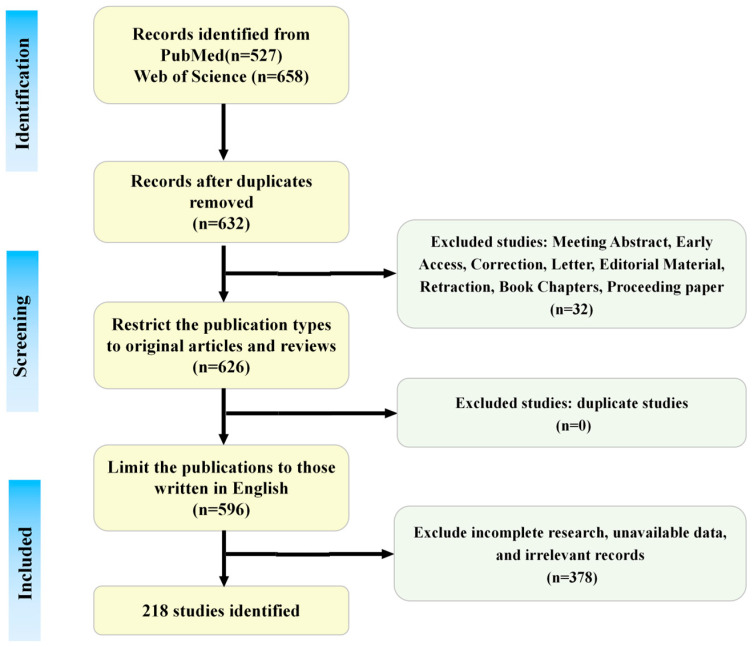
The literature search and screening flowchart. This figure is original and was created by the authors for this publication.

**Figure 3 biomolecules-15-01671-f003:**
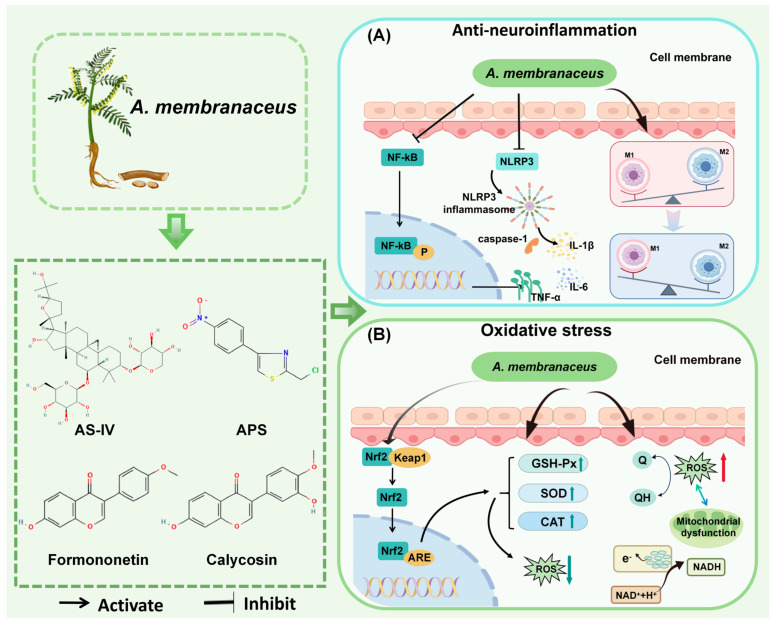
Mechanisms of *A. membranaceus* in Combating Neuroinflammation and Oxidative Stress. (**A**) Anti-neuroinflammation pathway. *A. membranaceus* components, notably Astragaloside IV (AS-IV) and *A. membranaceus* polysaccharides (APS), exert potent anti-inflammatory effects primarily by inhibiting the NF-κB signaling pathway. This inhibition results in a decrease in the production of key pro-inflammatory cytokines, including TNF-α, IL-1β, and IL-6. Furthermore, these bioactive compounds suppress the activation of the NLRP3 inflammasome, a critical complex that drives the maturation and secretion of IL-1β, thereby mitigating a central inflammatory cascade in CNS disorders. Concurrently, *A. membranaceus* promotes the polarization of microglia/macrophages from a pro-inflammatory M1 phenotype towards an anti-inflammatory and reparative M2 phenotype, contributing to the resolution of neuroinflammation. (**B**) Antioxidant stress pathway. *A. membranaceus* and its flavonoid constituents, such as Calycosin and Formononetin, activate the central antioxidant regulator Nrf2. Upon activation, Nrf2 translocates to the nucleus and binds to the Antioxidant Response Element (ARE), initiating the transcription of a battery of cytoprotective and antioxidant enzymes. This includes the upregulation of Superoxide Dismutase (SOD), Catalase (CAT), and Glutathione Peroxidase (GSH-Px). These enzymes work in concert to neutralize reactive oxygen species (ROS): SOD catalyzes the conversion of superoxide anions into hydrogen peroxide, which is then subsequently degraded to water (H_2_O) by CAT and GSH-Px. This enhanced antioxidant capacity effectively reduces oxidative damage to cellular components, a common feature in various CNS diseases. This figure is original and was created by the authors for this publication.

**Figure 4 biomolecules-15-01671-f004:**
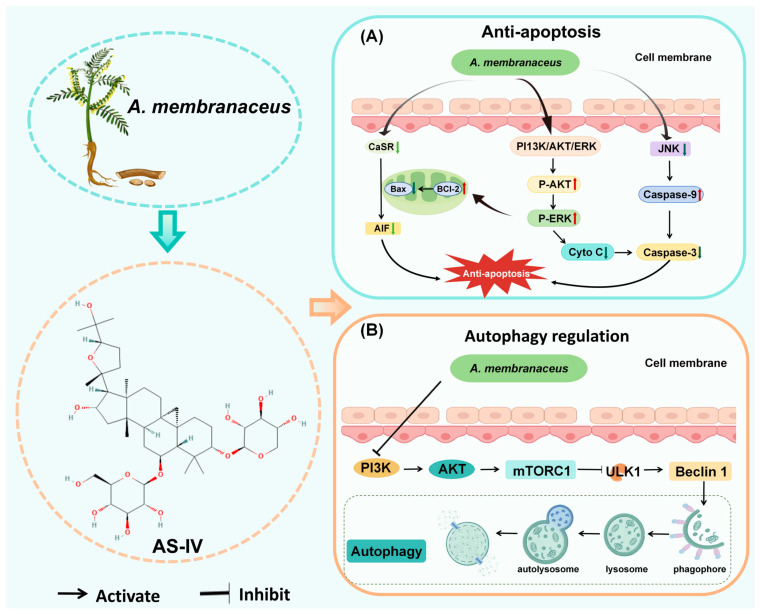
*A. membranaceus* exerts anti-apoptotic and autophagy regulatory effects. (**A**) Anti-apoptotic signaling pathways. AS-IV inhibits neuronal apoptosis through multiple pathways. AS-IV suppresses the phosphorylation of c-Jun N-terminal kinase (JNK) and downregulates the expression of the calcium-sensing receptor (CaSR), thereby preventing the activation of pro-apoptotic cascades. Concurrently, it activates the pro-survival PI3K/Akt and ERK signaling pathways. The net effect is a shift in the mitochondrial balance towards cell survival, characterized by a decrease in the Bax/Bcl-2 ratio, inhibition of cytochrome c (Cyto C) and apoptosis-inducing factor (AIF) release from mitochondria, and subsequent suppression of the key executioner caspase, caspase-3. (**B**) Regulation of autophagy. *A. membranaceus* components modulate the highly conserved autophagy pathway to exert neuroprotective effects. A key mechanism involves the inhibition of the PI3K/Akt/mTORC1 signaling axis. By reducing mTORC1 activity, *A. membranaceus* relieves its suppression on the ULK1 complex and Beclin-1, thereby initiating autophagosome formation. This promotes the sequestration of damaged organelles and protein aggregates into a phagophore, which matures into an autophagosome and subsequently fuses with a lysosome to form an autolysosome, where cargo is degraded and recycled. The regulation of this process helps maintain cellular homeostasis in neurons under stress. This figure is original and was created by the authors for this publication.

**Figure 5 biomolecules-15-01671-f005:**
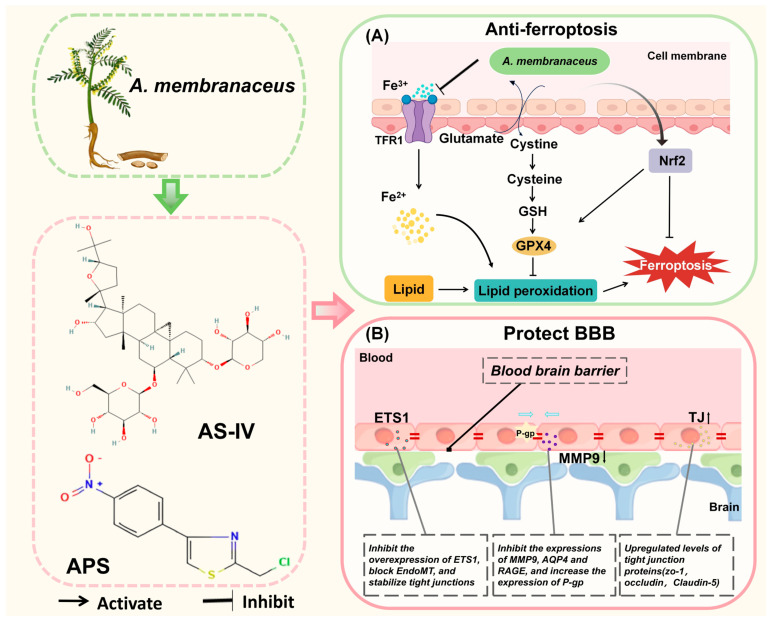
*A. membranaceus* exerts anti-ferroptosis and anti-blood–brain barrier damage effects. (**A**) Inhibition of ferroptosis. *A. membranaceus* and its bioactive compounds, such as total flavonoids (TFA) and AS-IV, combat ferroptosis, an iron-dependent form of cell death. They upregulate the system Xc-transporter (comprising the SLC7A11 subunit), which enhances cystine uptake and its reduction to cysteine for the synthesis of the key antioxidant glutathione (GSH). Concurrently, they activate the Nrf2 signaling pathway, which promotes the expression of glutathione peroxidase 4 (GPX4). GPX4 utilizes GSH to reduce toxic lipid hydroperoxides (L-OOH) to non-toxic lipid alcohols (L-OH), thereby halting the lipid peroxidation chain reaction that drives ferroptosis. Additionally, modulation of iron metabolism (e.g., via Transferrin Receptor 1, TFR1) may contribute to reducing the intracellular pool of reactive Fe^2+^. (**B**) Protection of the Blood–Brain Barrier (BBB). *A. membranaceus* components preserve BBB integrity through several concerted actions. Astragalus polysaccharides (APS) inhibit the transcription factor ETS1, thereby blocking Endothelial-to-Mesenchymal Transition (EndoMT) and helping to stabilize tight junctions (TJ). Key active ingredients, such as AS-IV and Cycloastragenol (CAG), upregulate the expression of tight junction proteins, including Zonula Occludens-1 (ZO-1), occludin, and Claudin-5, which seal the paracellular space. Furthermore, they protect the BBB by inhibiting the expression and activity of matrix metalloproteinase-9 (MMP-9) and Aquaporin-4 (AQP4), which are involved in basement membrane degradation and edema formation, respectively [103]. They also favorably modulate transporters by upregulating P-glycoprotein (P-gp) for efflux and downregulating the Receptor for Advanced Glycation End-products (RAGE) for reduced influx. This figure is original and was created by the authors for this publication.

**Figure 6 biomolecules-15-01671-f006:**
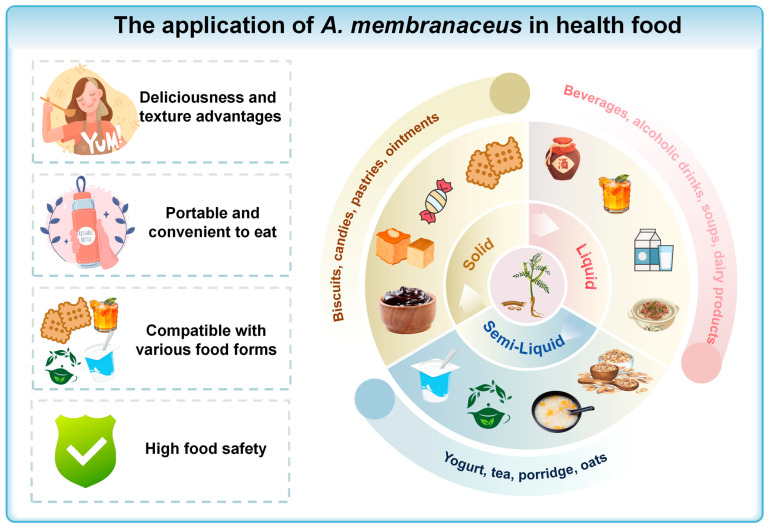
The application of *A. membranaceus* in health food. This figure is original and was created by the authors for this publication.

**Table 1 biomolecules-15-01671-t001:** Summary of Neuroprotective Mechanisms of Major Bioactive Compounds from *A.membranaceus*.

Disease	Bioactive Compound	Major Molecular Targets/Pathways	Related Disease Models	Refs
IS	AS-IV	PPARγ	tMCAO rat model	[14]
EAE	AS-IV	TLR4/Myd88/NF-kB signalling pathway	EAE mice model	[15]
CIRI	AS-IV	NLRP3	MCAO/R rat model	[16]
Depression	AS-IV	PPARγ/NF-kB/NLRP3	RRS-induced mice model of depression	[17]
TBI	AS-IV	PERK-eIF2α-ATF4 signaling pathway	a mouse TBI model	[18]
Brain death induced by anesthesia	AS-IV	NF-kB, JNK	Rat model induced by isoflurane	[19]
ischemia/reperfusion damages	AS-IV	Fas, FasL, Caspase-8, and Bax/Bcl-2	OGD/R rat model	[20]
AD	AS-IV	AMPK/mTOR	MCAO rat model	[21]
EBI	AS-IV	Nrf2/HO-1	MCAO rat model	[22]
I/R	AS-IV	MMP-9 and AQP4	I/R rat model	[23]
AIS	APS	P2X7R	MCAO rat model	[24]
EAE	APS	ETS1 and BBB	EAE mice model	[25]
AD	CAG	P-gp and RAGE	An immortalized endothelial cell line (bEnd.3)	[26]
SCI	CA	Hsp90-Akt/ASK1-p38 pathway	The vascular clamp compression SCI model	[27]
PD	TFA	SLC7A11/GPX-4 signaling pathway	MPTP/MPP-induced PD mouse model	[28]
AD	AST	Fas/Fasl-VDAC1	AD mouse models induced by Aβ1-42 and Aβ25-35	[29]
AD	AST	PI3K/Akt-mTOR-mediated autophagy	APP/PS1 mice	[30]

**Table 2 biomolecules-15-01671-t002:** Application of A.membranaceus in health food products and their potential relevance to CNS health.

Product Category	Representative Product	Health Claims & Potential Benefits	Proposed Mechanisms & Scientific Rationale
Liquid Products	*A. membranaceus* Chicken Soup (with black-boned chicken, goji berries)	Alleviates fatigue, improves sub-health (e.g., qi and blood deficiency), and may support appetite in chronic conditions (e.g., PD).	Nourishes qi and blood in TCM theory. Modern research suggests AM’s active compounds (e.g., AS-IV, APS) may combat fatigue and modulate energy metabolism. The formulation provides easily absorbable nutrients.
Semi-Liquid Products	*A. membranaceus* Porridge (with Japonica rice)	Aids post-illness/surgery recovery, may help alleviate symptoms like memory loss in AD patients with spleen-kidney deficiency.	Easy to digest and absorb. *A. membranaceus* fortifies the spleen and kidney in TCM, which are considered fundamental to cognitive function. Provides stable energy release.
Solid Products	*A. membranaceus* Health Biscuits	Provides convenient nutritional support, enhances physical well-being for patients with CNS disorders.	Portable and palatable. Offers a functional snack option. The potential systemic anti-inflammatory and antioxidant effects of *A. membranaceus* may indirectly support brain health.

## Data Availability

The data used in this article were sourced from materials mentioned in the References Section.

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
