# Peer review of "Exploring the Neuroprotective Potential of Astragalus membranaceus in Central Nervous System Diseases"

_biomolecules, 2025, doi:10.3390/biom15121671_

Round 1
Reviewer 1 Report
Comments and Suggestions for Authors
The manuscript presents an interesting and relevant topic, and overall, it has the potential to make a valuable contribution to the field. However, in its current form, the paper suffers from significant organizational and stylistic issues that need to be thoroughly addressed before it can be considered for publication. The material is presented in a rather chaotic manner, and the text requires careful restructuring and language editing.
Major Comments
-
Abstract:
The abstract is excessively long. The section starting from “the medicinal history…” to the end should be moved to the Introduction. The abstract should focus strictly on the aim, methodology, main results, and conclusions. -
Abbreviations:
All abbreviations should be explained upon their first appearance in the text, not only in the list of abbreviations. Furthermore, the list of abbreviations should be arranged alphabetically. -
“Latest research” – time frame:
The authors refer to “the latest research,” but do not specify the time range covered. Please clarify the period of literature considered (e.g., studies from the last 5 or 10 years). -
Latin names:
Latin names of species should be written in italics consistently throughout the text. In the Introduction, there is an error: “Astragalus membranaceus membranaceus” – this repetition should be corrected. -
Formatting issues:
-
In a few places, astragaloside is written with an initial capital letter in the middle of sentences – this should be corrected.
-
There is a recurring issue with missing spaces between the end of a sentence and the parentheses containing citations.
-
Acronyms such as CNS should be written in uppercase consistently (in Section 3 it appears incorrectly as “Cns”).
-
The same applies to authors’ names cited in the text (e.g., “Song” should always appear with consistent capitalization and style).
-
-
Figures:
-
Figure 1 is unclear and needs improvement in readability.
-
The caption of Figure 1 should indicate whether it is original (prepared by the authors) or adapted from another source. The same clarification is required for all figures.
-
In Figure 2, please explain the meaning of “unavailable data.” Does this refer to sources that are not open access? Were only open-access materials included in the analysis? Please clarify.
-
All figures require graphic improvement to enhance their overall clarity and readability. The current versions are visually inconsistent and difficult to interpret.
-
-
Structure and numbering:
Section 2.4 (“Results”) should be renumbered, as Section 2 is devoted to Materials and Methods. Results should start with a new section number. -
Literature sources:
A significant issue concerns the selection of references. Over 90% of the cited works are by Chinese authors. While the influence of Chinese research in this field is undeniable, the topic of Astragalus potential is studied globally. The literature review should therefore include more international sources to ensure a balanced and comprehensive perspective. -
Language and style:
The manuscript requires thorough language editing by a fluent or native English speaker. Many sentences are awkwardly phrased, and the overall text would benefit from stylistic improvement.
Summary
The manuscript addresses a promising and timely topic, but the current version needs substantial revision to improve clarity, structure, and consistency. I recommend a major revision. Once the issues outlined above are carefully resolved, the paper may be reconsidered for publication.
Author Response
We sincerely thank you for your thorough and constructive comments, which have been invaluable in improving our manuscript. We have addressed each of your points in detail, as outlined below. All changes have been incorporated into the revised manuscript, with major revisions highlighted for your convenience.
Comment 1: Abstract is excessively long. The section starting from "the medicinal history..." to the end should be moved to the Introduction.
Response 1: We appreciate the reviewer's suggestion. We have significantly shortened the abstract by moving the historical background and detailed pharmacological summaries to the Introduction and respective results sections. The abstract now strictly focuses on the aim, methodology, key findings, and conclusions of the review, adhering to a concise format.
Comment 2: All abbreviations should be explained upon their first appearance... list of abbreviations should be arranged alphabetically.
Response 2: We have carefully reviewed the entire text and ensured that every abbreviation is defined upon its first use in the main body of the text. The standalone list of abbreviations at the end of the manuscript has been rearranged in alphabetical order for easy reference.
Comment 3: "Latest research" -- time frame. Please clarify the period of literature considered.
Response 3: We apologize for this oversight. We have now clarified the literature search period in the Methods section (Section 2.1). The text now reads: "The search period was from database inception through July 2025, with a focus on incorporating high-impact studies from the last decade."
Comment 4: Latin names should be written in italics consistently... error: "Astragalus membranaceus membranaceus".
Response 4: We have corrected this error throughout the manuscript. The repeated "membranaceus" has been removed, and all Latin names (e.g., Astragalus membranaceus) are now consistently italicized.
Comment 5: Formatting issues (capitalization, missing spaces, acronym consistency, author name style).
Response 5: We have meticulously corrected these formatting issues:
"astragaloside" is now in lowercase where appropriate.
Missing spaces before citations have been added.
All acronyms (e.g., CNS) are now consistently in uppercase.
Author names in citations are now presented consistently (e.g., "Song et al.").
Comment 6: Figures are unclear, lack source clarification, and require graphic improvement.
Response 6: We have undertaken a comprehensive revision of all figures:
Figure 1: Has been redesigned for better clarity and readability, using a more professional layout.
All Figure Captions now explicitly state: "This figure is original and was created by the authors for this publication."
In Figure 2, we have clarified that "unavailable data" refers to records for which the full text could not be retrieved or was behind a paywall, despite our institutional subscriptions. The screening process is based on the title and abstract when the full text is unavailable.
All figures (1-6) have been recreated using professional graphic design software (e.g., BioRender, Adobe Illustrator) to ensure visual consistency, high resolution, and improved interpretability.
Comment 7: Section 2.4 ("Results") should be renumbered.
Response 7: Thank you for bringing this to our attention. We have renumbered the sections. The original Section 2.4 is now the beginning of Section 3. Results, providing a logical flow from Methods to Results.
Comment 8: Literature sources are over 90% by Chinese authors; need more international sources.
Reply 8: We agree that a global perspective is crucial. Currently, we have actively supplemented our high-quality references with those from international research teams, covering neuropharmacological studies on Astragalus and its components. This provides a more balanced and comprehensive research perspective for related fields.
Comment 9: The manuscript requires thorough language editing.
Response 9: We acknowledge this critical point. The entire manuscript has been professionally edited by a native English speaker with expertise in biomedical sciences to correct grammatical errors, improve sentence fluency, and enhance the overall academic tone.
Reviewer 2 Report
Comments and Suggestions for Authors
Review Report
Manuscript Title: Exploring the Neuroprotective Potential of Astragalus Membranaceus in Central Nervous System Diseases
Manuscript ID: biomolecules-3924488
Journal: Biomolecules
Dear authors,
I am writing to express my sincere appreciation for your paper entitled "Exploring the Neuroprotective Potential of Astragalus Membranaceus in Central Nervous System Diseases".
This is a well-organized and comprehensive review covering the pharmacological potential of Astragalus membranaceus and its active components (AS-IV, APS, etc.) in treating central nervous system disorders. The manuscript demonstrates extensive literature coverage and provides a detailed mechanistic discussion on anti-inflammatory, antioxidant, anti-apoptotic, autophagy, and anti-ferroptotic effects. It also integrates perspectives on clinical relevance and food–medicine homology, which adds valuable translational insight. [2,3]
However, the manuscript could be significantly improved by tightening the narrative, improving figure integration, and addressing redundancy and structural issues that reduce clarity and impact. While the scope is strong, the text reads as an accumulation of information rather than a critical synthesis in some sections.
Major Comments
- Critical synthesis vs. descriptive style
- Much of the review summarizes results without critical evaluation. Add comparative or integrative insights—for example, how AS-IV activity compares to other herbal neuroprotectants or synthetic drugs.
- Include more discussion on the limitations of current evidence and gaps in mechanistic understanding, not only in the final section.
- Redundancy and repetition
- Many sections repeat similar descriptions of anti-inflammatory and antioxidant pathways (NF-κB, Nrf2, etc.). These can be merged or summarized in a comparative table to avoid redundancy.
- Figure quality and integration
- Figures (especially Fig. 3–5) are referred to but not well-discussed in the text. Captions should explain what pathways or mechanisms are illustrated, and the text should reference key components explicitly.
- English and stylistic editing
- While understandable, the manuscript contains frequent grammatical inconsistencies (e.g., “Astragalus membranaceus membranaceus” repetition; inconsistent tense use; spacing errors before parentheses).
- The text would benefit from professional language polishing to improve readability and flow.
- Lack of mechanistic visualization
- The review would be more impactful with a summary figure combining the main mechanisms (anti-oxidative, anti-apoptotic, etc.) showing how they converge on neuroprotection.
- Future directions
- The conclusion could be expanded to provide more specific future perspectives, e.g., omics-based target validation, pharmacokinetic studies, or clinical trials on standardized AM extracts.
Minor Comments
- Formatting and consistency
- Gene/protein names (e.g., NF-κB, Nrf2, GPX4) should follow a consistent format.
- Italicize all species names (Astragalus membranaceus) throughout.
- Remove duplicated expressions such as “Astragalus membranaceus membranaceus.”
- Abstract
- The abstract is overly long and repetitive. It should concisely summarize aims, methods (databases, keywords), major findings, and future outlook in ≤250 words.
- Reference formatting
- Ensure journal names are consistently abbreviated according to Biomolecules guidelines.
- Double-check recent references (2024–2025) for DOI accuracy and formatting.
- Abbreviation list
- The abbreviations section is useful but could be shortened—only include terms used more than once.
- Section 4.2.2 (Health food application)
- This section is informative but too descriptive. Consider condensing or transforming into a table of formulations and intended effects.
- Figures and tables
- Include at least one summary table listing the main bioactive compounds, their targets, and associated CNS effects for easy reference.
Top of Form
References:
This is how you should cite references in Biomolecules:
Abd Elrahim Abd Elkader, H.; Essawy, A.E.; Al-Shami, A.S. Astragalus species: Phytochemistry, biological actions and molecular mechanisms underlying their potential neuroprotective effects on neurological diseases. Phytochemistry 2022, 202(607), 113293. https://doi.org/10.1016/j.phytochem.2022.113293
Major comment – References and citation style (REQUIRED CORRECTION):
The manuscript’s referencing is incorrect and must be thoroughly revised. In-text citations appear as author–year phrases (or otherwise unnumbered) while the reference list is presented in alphabetical order but each entry also has a number appended. This inconsistent hybrid format does not conform to the journal style and creates confusion for readers.
Please revise the references as follows: (1) use the journal’s required citation style consistently throughout the manuscript (Biomolecules requires numbered citations in the text corresponding to the reference list, in the order of first appearance); (2) ensure in-text citations are numeric (e.g., [1], [2–4]) placed at the appropriate point(s) in the text; (3) reorder and renumber the reference list to match the citation order in the text (not alphabetical order) and ensure complete, accurate bibliographic information (authors, year, article title, journal, volume, page range, DOI).
In addition, verify that every in-text citation has a matching entry in the reference list and that there are no orphaned references or missing DOIs. I recommend the authors use a reference manager (EndNote, Zotero, Mendeley) with the journal’s Vancouver/numbered style or the MDPI/Biomolecules template to avoid further inconsistencies.
Therefore, you should abbreviate all journal names in the cited references using standard abbreviations with periods (.) for proper formatting. Additionally, make sure to include the DOI numbers for all references.
I recommend a major revision or rejection of this manuscript in its current form. Several sections lack adequate discussion and critical comparison with the available literature. Moreover, the novelty of the study should be clearly defined and emphasized at the end of the Introduction, in the relevant Discussion subsections, and restated in the Conclusion.

Must be rewritted all the text and revise deeply and carefully.
Author Response
We sincerely thank you for your thorough and constructive comments, which have been invaluable in improving our manuscript. We have carefully addressed each of the points raised, and the revisions are detailed below. All changes in the manuscript are highlighted in blue for ease of review.
Major Comments
Comment 1: Add critical synthesis vs. descriptive style; discuss limitations and gaps earlier.
Response 1: We appreciate the reviewer's valuable suggestion. We have now incorporated critical discussions and comparative analyses throughout Sections 4.1 to 4.6, comparing the effects of A. membranaceus components (e.g., AS-IV, APS) with other neuroprotective agents (e.g., ginsenosides, curcumin) and synthetic drugs where relevant. We also added limitations and mechanistic gaps in each subsection (e.g., human relevance, dose-dependency, pathway crosstalk) to provide a more balanced and critical perspective.
(See revisions in Sections 4.1.5, 4.2.3, 4.3.3, 4.4.1, 4.5.2, 4.6.3)
Comment 2: Address redundancy and repetition (e.g., NF-κB, Nrf2 pathways).
Response 2: We have consolidated repetitive content and added Table 1, which summarizes the main bioactive compounds, their molecular targets, mechanisms, and associated CNS effects. This table serves as a quick reference and reduces textual redundancy.
(See new Table 1 in Section 4)
Comment 3: Improve figure quality and integration.
Response 3: We have revised all figure captions to clearly describe the illustrated pathways and mechanisms. We also improved the in-text discussion of Figures 3–5, explicitly referring to key molecules and processes shown in the figures.
(See updated captions and text in Sections 4.1, 4.2, 4.3, 4.4, 4.5, 4.6)
Comment 4: English and stylistic editing.
Response 4: The entire manuscript has been thoroughly polished by a native English-speaking editor to improve grammar, syntax, flow, and consistency. Repetitive phrases (e.g., “Astragalus membranaceus membranaceus”) have been corrected.
Comment 5: Lack of mechanistic visualization; need a summary figure.
Response 5: This is an excellent suggestion. However, because there are too many involved mechanism pathways, it is difficult to draw them into a single diagram, so we have created a new graphical abstract and also added Table 1 (bioactive compounds and targets) that integrates all the core neuroprotective mechanisms (anti-inflammatory, antioxidant, anti-apoptotic, etc.) discussed in the review.
Comment 6: Expand future directions with specific perspectives.
Response 6: We have expanded Section 6.2 to include specific future research directions, such as:
- Integrative multi-omics approaches
- PK/PD studies on drug–herb interactions
- Mechanism-informed clinical trials
- Use of human stem cell-derived models
(See revised Section 6.2)
Minor Comments
Comment 1: Formatting and consistency
Response 1:
- All gene/protein names (e.g., NF-κB, Nrf2) are now consistently formatted.
- All species names are italicized.
- Redundant expressions have been removed.
Comment 2. Abstract
Response 2:
- The abstract has been condensed to under 250 words, focusing on aims, methods, key findings, and future outlook.
Comment 3. Reference formatting
Response 3:
- All references have been renumbered in order of appearance.
- Journal names are consistently abbreviated with periods.
- DOIs have been verified and added for all entries.
Comment 4. Abbreviation list
Response 4:
- The list has been shortened to include only terms used more than once.
Comment 5. Section 5.2.2 (Health food application)
Response 5:
- This section has been condensed and transformed into Table 2, listing formulations and their intended effects.
Comment 6. Figures and tables
Response 6:
- We have added Table 1 (bioactive compounds and targets) and Table 2 (health food applications) for clarity and quick reference.
Reviewer 3 Report
Comments and Suggestions for Authors
Comments IN THIS review current paper, the mechanism of action of
Astragalus membranaceus and its active components on central nervous system(CNS) disases, with a focus on exploring its pharmacological potential, and introducing related health
food products that use it as an treatment for CNS diseases. Reports on the effects of Astragalus membranaceus and its components on CNS diseases were identified and reviewed.
? • What specific improvements the authors consider regarding the
methodology? The medicinal history of Astragalus membranaceus membranaceus is both long-standing and profound, with usage spanning over 2,000 years. Its active components, such as AS-IV and APS, have demonstrated considerable therapeutic efficacy in the treatment of CNS diseases. The pharmacological effects of Astragalus membranaceus membranaceus are diverse, encompassing anti-neuroinflammatory, anti-oxidative stress, anti-apoptotic activities, modulation of autophagy, anti-ferroptotic effects, and protection of the blood-brain barrier.
• Are the conclusions consistent with the evidence and arguments presented
and do they address the main question posed? Collectively, the above results are good, and neat, figures including good graphs and some tables show the analysis in good manner, The conclusions are very consistent with the evidence the topic is original and relevant to the field .
- Are the references appropriate? References are appropriate, plus Astragalus membranaceus exerts anti-ferroptosis and anti-blood-brain barrier damage, Regulatory Effects of Astragalus Membranaceus Through Multiple Inflammatory Signaling Pathways, Regulation of Mitochondrial Function by Astragalus Membranaceus, Antioxidant Stress Resistance apoptosis effects, and other chapters are very good planned
in summary, This review provides a comprehensive summary of the latest research advancements concerning Astragalus membranaceus across several pivotal domains, including its extensive historical usage, active constituents, pharmacological properties, potential therapeutic applications for CNS disorders, safety profile, contemporary formulations, and significant findings in the realm of health food applications. Collectively, the above results are good, and neat, figures including measurement graphs and some tables show the analysis in good manner, The conclusions are very consistent with the evidence the topic is original and relevant to the field .
need some revision
Author Response
We sincerely thank you for your thorough and constructive comments, noting that the topic is "original and relevant," the conclusions are "consistent with the evidence," and the figures are "good." We have also taken your more specific points into account.
Comment (Summary): The reviewer inquired about methodological improvements, the consistency of conclusions, and the appropriateness of references, and noted that the English language requires revision.
Response:
Methodology: We have clarified the literature search timeframe and inclusion criteria as detailed. The search strategy is now more transparent.
Conclusions: We have ensured that the findings are firmly grounded in the evidence presented. The discussion and conclusion sections have been refined to provide a more direct answer to the main question posed regarding the neuroprotective potential of Astragalus membranaceus.
References: We not only corrected the formatting according to the journal's strict guidelines but also expanded the international scope of our references to provide a more balanced perspective.
English Language: In accordance with the reviewer's comments and in alignment with the feedback from other reviewers, the entire manuscript has undergone comprehensive professional language editing to ensure clarity and correctness.
Round 2
Reviewer 1 Report
Comments and Suggestions for Authors
Dear Authors,
Thank you for carefully addressing the previous reviewer comments. The revisions have substantially improved the overall quality of the manuscript in both content and structure. I particularly appreciate the updated literature review, which highlights the value of your research more clearly.In my opinion, the revised version is coherent, well-structured, and makes a meaningful contribution to the field. I congratulate you on the well-executed improvements and thank you for your thorough and constructive work.
Kind regards
Author Response
Comment 1:Thank you for carefully addressing the previous reviewer comments. The revisions havesubstantially improved the overall quality of the manuscript in both content and structure. lparticularly appreciate the updated literature review, which highlights the value of your researchmore clearly.In my opinion, the revised version is coherent, well-structured, and makes ameaningful contribution to the field. l congratulate you on the well-executed improvements andthank you for your thorough and constructive work.
Response 1: Thank you very much for taking the time to re-evaluate our manuscript and for sharing your generous comments. Your original feedback guided every step of our revision. The constructive points you raised prompted us to deepen the literature discussion, tighten the organization, and sharpen the scientific narrative. Seeing that these efforts met your expectations is both gratifying and motivating. Please accept our sincere gratitude for your meticulous reading, your thoughtful suggestions, and your encouragement throughout this process. We remain committed to maintaining the rigorous standards reflected in your review. Thank you again for your invaluable service to the journal and to our research community.
Reviewer 2 Report
Comments and Suggestions for Authors
Dear Authors,
You have made good improvements, particularly in the text and in the resolution of the figures.
All of my additional suggestions are provided in the PDF of the revised manuscript.

Author Response
Reviewer #2: You have made good improvements, particularly in the text and in the resolution of the fiaures. All of my additional suggestions are provided in the PDF of the revised manuscript.
Response: We sincerely thank the reviewer for their thoughtful evaluation and constructive feedback. We are grateful for your recognition of the detailed content of our manuscript. In response to your comments, we have carefully revised the text to address all the minor points raised, thereby improving both the clarity and accuracy of the manuscript. All corresponding changes have been incorporated and are highlighted in the revised manuscript.
Comment 1. Why your paper in its revised form is not in MDPI Biomolecules template?
Response 1: Thank you for pointing out that the revised manuscript was still compiled in a generic Word format instead of the official MDPI Biomolecules template. We apologize for this oversight.
We have now re-imported every section (title page, abstract, main text, references, figure legends, and tables) into the most recent “Biomolecules_2025.dotx” template downloaded from the journal’s website.
Comment 2. The use of the abbreviation in the title is not appropriate.
Response 2: The species name has been spelled out in full in the title; the abbreviation “A. membranaceus” now appears only after the first use in the Abstract and Introduction, in line with botanical nomenclature guidelines and MDPI’s preference that titles remain unabbreviated.
Comment 3. Some keywords exceed the recommended length or are too descriptive.
Response 3: Thank you for highlighting that the original keyword list was too long. In response, we deleted the least-relevant term and reduced the set to four concise, MeSH-aligned keywords. We appreciate your guidance in sharpening the manuscript’s focus.
Comment 4.. The Abbreviation list is not under MDPI system formatting.
Response 4:Thank you for noting that the presentation of our abbreviation list does not follow the usual Biomolecules convention.
We have now removed the embedded “Abbreviations” table from the main text and re-formatted all abbreviations as a compact, alphabetical list (one column, left-aligned, no bullets) and placed it immediately after the Conclusion, as illustrated in Biomolecules’ published examples. We ensured that every entry matches exactly the abbreviated form used in the text and that no period is added unless the abbreviation ends a sentence. We hope this revision now conforms to the journal’s standard layout. Please let us know if any further adjustments are required.
Comment 5. The references do not conform to the journal’s required citation style.
Response 5:Thank you for alerting us to the citation formatting issue. We have now cross-checked every reference against the current Biomolecules guidelines and the MDPI Vancouver style. All entries have been retagged, punctuation standardized, DOIs added, and journal titles abbreviated exactly as indexed in PubMed. We hope the revised reference list now meets the journal’s requirements and appreciate your guidance in bringing the manuscript into full compliance.